# ENHANCING THE TRANSFORMER WITH EXPLICIT RELATIONAL ENCODING FOR MATH PROBLEM SOLVING

## ABSTRACT

We incorporate Tensor-Product Representations within the Transformer in order to better support the explicit representation of relation structure. Our Tensor-Product Transformer (TP-Transformer) sets a new state of the art on the recently-introduced Mathematics Dataset containing 56 categories of free-form math word-problems. The essential component of the model is a novel attention mechanism, called TP-Attention, which explicitly encodes the relations between each Transformer cell and the other cells from which values have been retrieved by attention. TP-Attention goes beyond linear combination of retrieved values, strengthening representation-building and resolving ambiguities introduced by multiple layers of standard attention. The TP-Transformer's attention maps give better insights into how it is capable of solving the Mathematics Dataset's challenging problems. Pretrained models and code will be made available after publication.

## 1 INTRODUCTION

In this paper we propose a variation of the Transformer (Vaswani et al., 2017) that is designed to allow it to better incorporate structure into its representations. We test the proposal on a task where structured representations are expected to be particularly helpful: math word-problem solving, where, among other things, correctly parsing expressions and compositionally evaluating them is crucial. Given as input a free-form math question in the form of a character sequence like `Let r(g) be the second derivative of 2*g**3/3 - 21*g**2/2 + 10*g. Let z be r(7). Factor -z*s + 6 - 9*s**2 + 0*s + 6*s**2.`, the model must produce an answer matching the specified target character-sequence `-(s + 3)*(3*s - 2)` exactly. Our proposed model is trained end-to-end and infers the correct answer for novel examples without any task-specific structural biases.

We begin by viewing the Transformer as a kind of Graph Neural Network (e.g., Gori et al., 2005). For concreteness, consider the encoder component of a Transformer with $H$ heads. When the $h^{\text{th}}$ head of a cell $t$ of layer $l$ issues a query and as a result concentrates its self-attention distribution on another cell $t'$ in layer $l$, we can view these two cells as joined by an edge in an information-flow graph: the information content at $t'$ in effect passes via this edge to affect the state of $t$. The strength of this attention can be viewed as a weight on this edge, and the index $h$ of the head can be viewed as a label. Thus, each layer of the Transformer can be viewed as a complete, directed, weighted, labeled graph. Prior NLP work has interpreted certain edges of these graphs in terms of linguistic relations (Sec. 7), and we wish to enrich the relation structure of these graphs to better support the explicit representation of relations within the Transformer.

Here we propose to replace each of the discrete edge labels $1, \ldots, H$, with a **relation vector**: we create a bona fide representational space for the relations being learned by the Transformer. This makes it possible for the hidden representation at each cell to approximate the vector embedding of a symbolic structure built from the relations generated by that cell. This embedding is a **Tensor-Product Representation** (**TPR**; Smolensky, 1990) in an end-to-end-differentiable TPR system (Schlag & Schmidhuber, 2018; Schmidhuber, 1992) that learns "internal spotlights of attention" (Schmidhuber, 1993). TPRs provide a general method for embedding symbol structures in vector spaces. TPRs support compositional processing by directly encoding constituent structure: the representation of a structure is the sum of the representation of its constituents. The representation of each constituent is built compositionally from two vectors: one vector that embeds the content of the constituent, the

'**filler**' — here, the vector returned by attention — and a second vector that embeds the structural **role** it fills — here, a relation conceptually labeling an edge of the attention graph. The vector that embeds a filler and the vector that embeds the role it fills are **bound** together by the tensor product to form the tensor that embeds the constituent that they together define.[1] The relations here, and the structures they define, are learned unsupervised by the Transformer in service of a task; post-hoc analysis is then required to interpret those roles.

In the new model, the **TP-Transformer**, each head of each cell generates a key-, value- and query-vector, as in the Transformer, but additionally generates a **role-vector** (which we refer to in some contexts as a 'relation vector'). The query is interpreted as seeking the appropriate filler for that role (or equivalently, the appropriate string-location for fulfilling that relation). Each head binds that filler to its role via the tensor product (or some contraction of it), and these filler/role bindings are summed to form the TPR of a structure with $H$ constituents (details in Sec. 2).

An interpretation of an actual learned relation illustrates this (see Fig. 4 in Sec. 5.2). One head of our trained model can be interpreted as partially encoding the relation *second-argument-of*. The top-layer cell dominating an input digit seeks the operator of which the digit is in the second-argument role. That cell generates a vector $r_t$ signifying this relation, and retrieves a value vector $v_{t'}$ describing the operator from position $t'$ that stands in this relation. The result of this head's attention is then the binding of filler $v_{t'}$ to role $r_t$; this binding is added to the bindings resulting from the cell's other attention heads.

On the Mathematics Dataset (Sec. 3), the new model sets a new state of the art for the overall accuracy (Sec. 4), and for all the individual-problem-type module accuracies (Fig. 2). Initial results of interpreting the learned roles for the arithmetic-problem module show that they include a good approximation to the second-argument role of the division operator and that they distinguish between numbers in the numerator and denominator roles (Sec. 5).

More generally, it is shown that Multi-Head Attention layers not only capture a subspace of the attended cell but capture nearly the full information content (Sec. 6.1). An argument is provided that multiple layers of standard attention suffer from the binding problem, and it is shown theoretically how the proposed TP-Attention avoids such ambiguity (Sec. 6.2). The paper closes with a discussion of related work (Sec. 7) and a conclusion (Sec. 8).

## 2    THE TP-TRANSFORMER

The TP-Transformer's encoder network, like the Transformer's encoder (Vaswani et al., 2017), can be described as a 2-dimensional lattice of cells $(t, l)$ where $t = 1, ..., T$ are the sequence elements of the input and $l = 1, ..., L$ are the layer indices with $l = 0$ as the embedding layer. All cells share the same topology and the cells of the same layer share the same weights. More specifically, each cell consists of an initial layer normalization (LN) followed by a **TP-Multi-Head Attention (TPMHA)** sub-layer followed by a fully-connected feed-forward (FF) sub-layer. Each sub-layer is followed by layer normalization (LN) and by a residual connection (as in the original Transformer; Eq. 1). Our cell structure follows directly from the official TensorFlow source code by Vaswani et al. (2017) but with regular Multi-Head Attention replaced by our TPMHA layer.

The input into $\text{cell}_{t,l}$ is the output of $\text{cell}_{t,l-1}$ and doesn't depend on the state of any other cells of the same layer, which allows a layer's outputs to be computed in parallel.

$$
\begin{aligned}
\boldsymbol{h}_{t,l} &= \boldsymbol{z}_{t,l} + \text{TPMHA}(\text{LN}(\boldsymbol{z}_{t,l}), \text{LN}(\boldsymbol{z}_{1:T,l})) \\
\boldsymbol{z}_{t,l+1} &= \text{LN}(\boldsymbol{h}_{t,l} + \text{FF}(\text{LN}(\boldsymbol{h}_{t,l})))
\end{aligned}
\tag{1}
$$

We represent the symbols of the input string as one-hot vectors $\boldsymbol{x}_1, ..., \boldsymbol{x}_T \in \mathbb{R}^{d_v}$ where $d_v$ is the size of the vocabulary and the respective columns of the matrix $\boldsymbol{E} \in \mathbb{R}^{d_z \times d_v}$ are the embedding vectors of those symbols. We also include a positional representation $\boldsymbol{p}_t$ using the same sinusoidal

---

[1] The tensor product operation (when the role-embedding vectors are linearly independent) enables the sum of constituents representing the structure as a whole to be uniquely decomposable back into individual pairs of roles and their fillers, if necessary.

encoding schema introduced by Vaswani et al. (2017). The input of the first-layer $\text{cell}_{t,1}$ is $\boldsymbol{z}_{t,0}$:

$$\begin{aligned}
\boldsymbol{e}_t &= \boldsymbol{E}\boldsymbol{x}_t\sqrt{d_z} + \boldsymbol{p}_t \\
\boldsymbol{r}_t &= \boldsymbol{W}^{(p)}\boldsymbol{e}_t + \boldsymbol{b}^{(p)} \\
\boldsymbol{z}_{t,0} &= \boldsymbol{e}_t \odot \boldsymbol{r}_t
\end{aligned} \tag{2}$$

where $\boldsymbol{W}^{(p)} \in \mathbb{R}^{d_z \times d_z}$, $\boldsymbol{b}^{(p)} \in \mathbb{R}^{d_z}$, $\boldsymbol{r}_t$ is a position- and symbol-dependent role representation, and $\odot$ is elementwise multiplication (a contraction of the tensor product: see Sec. 2.1).

## 2.1 TP-MULTI-HEAD ATTENTION

The TPMHA layer of the encoder consists of $H$ heads that can be applied in parallel. Every head $h, 1 \leq h \leq H$ applies separate affine transformations $\boldsymbol{W}_l^{h,(k)}, \boldsymbol{W}_l^{h,(v)}, \boldsymbol{W}_l^{h,(q)}, \boldsymbol{W}_l^{h,(r)} \in \mathbb{R}^{d_k \times d_z}$, $\boldsymbol{b}_l^{h,(k)}, \boldsymbol{b}_l^{h,(v)}, \boldsymbol{b}_l^{h,(q)}, \boldsymbol{b}_l^{h,(r)} \in \mathbb{R}^{d_k}$ to produce key, value, query, and relation vectors from the hidden state $\boldsymbol{z}_{t,l}$, where $d_k = d_z/H$:

$$\begin{aligned}
\boldsymbol{k}_{t,l}^h &= \boldsymbol{W}_l^{h,(k)}\boldsymbol{z}_{t,l} + \boldsymbol{b}_l^{h,(k)} & \boldsymbol{q}_{t,l}^h &= \boldsymbol{W}_l^{h,(q)}\boldsymbol{z}_{t,l} + \boldsymbol{b}_l^{h,(q)} \\
\boldsymbol{v}_{t,l}^h &= \boldsymbol{W}_l^{h,(v)}\boldsymbol{z}_{t,l} + \boldsymbol{b}_l^{h,(v)} & \boldsymbol{r}_{t,l}^h &= \boldsymbol{W}_l^{h,(r)}\boldsymbol{z}_{t,l} + \boldsymbol{b}_l^{h,(r)}
\end{aligned} \tag{3}$$

The filler of the attention head $t, l, h$ is

$$\bar{\boldsymbol{v}}_{t,l}^h = \sum_{i=1}^T \boldsymbol{v}_{i,l}^h \alpha_{t,l}^{h,i}, \tag{4}$$

i.e., a weighted sum of all $T$ values of the same layer and attention head (see Fig. 1). Here $\alpha_{t,l}^{h,i} \in (0,1)$ is a continuous *degree of match* given by the softmax of the dot product between the query vector at position $t$ and the key vector at position $i$:

$$\alpha_{t,l}^{h,i} = \frac{\exp(\boldsymbol{q}_{t,l}^h \cdot \boldsymbol{k}_{i,l}^h \frac{1}{\sqrt{d_k}})}{\sum_{i'=1}^T \exp(\boldsymbol{q}_{t,l}^h \cdot \boldsymbol{k}_{i',l}^h \frac{1}{\sqrt{d_k}})} \tag{5}$$

The scale factor $\frac{1}{\sqrt{d_k}}$ can be motivated as a variance-reducing factor under the assumption that the elements of $\boldsymbol{q}_{t,l}^h$ and $\boldsymbol{k}_{t,l}^h$ are uncorrelated variables with mean 0 and variance 1, in order to initially keep the values of the softmax in a region with better gradients.

Finally, we bind the filler $\bar{\boldsymbol{v}}_{t,l}^h$ with our relation vector $\boldsymbol{r}_{t,l}^h$, followed by an affine transformation $\boldsymbol{W}_{h,l}^{(o)} \in \mathbb{R}^{d_z \times d_k}, \boldsymbol{b}_{h,l}^{(o)} \in \mathbb{R}^{d_z}$ before it is summed up with the other heads' bindings to form the TPR of a structure with $H$ constituents: this is the output of the TPMHA layer.

$$\text{TPMHA}(\boldsymbol{z}_{t,l}, \boldsymbol{z}_{1:T,l}) = \sum_h \left[ \boldsymbol{W}_{h,l}^{(o)}(\bar{\boldsymbol{v}}_{t,l}^h \odot \boldsymbol{r}_{t,l}^h) + \boldsymbol{b}_{h,l}^{(o)} \right] \tag{6}$$

Note that, in this binding, to control dimensionality, we use a contraction of the tensor product, pointwise multiplication $\odot$: this is the diagonal of the tensor product. For discussion, see the Appendix.

It is worth noting that the $l^{\text{th}}$ TPMHA layer returns a vector that is quadratic in the inputs $\boldsymbol{z}_{t,l}$ to the layer: the vectors $\boldsymbol{v}_{i,l}^h$ that are linearly combined to form $\bar{\boldsymbol{v}}_{t,l}^h$ (Eq. 4), and $\boldsymbol{r}_{t,l}^h$, are both linear in the $\boldsymbol{z}_{i,l}$ (Eq. 3), and they are multiplied together to form the output of TPMHA (Eq. 6). This means that, unlike regular attention, TPMHA can increase, over successive layers, the polynomial degree of its representations as a function of the original input to the Transformer. Although it is true that the feed-forward layer following attention (Sec. 2.2) introduces its own non-linearity even in the standard Transformer, in the TP-Transformer the attention mechanism itself goes beyond mere linear re-combination of vectors from the previous layer. This provides further potential for the construction of increasingly abstract representations in higher layers.

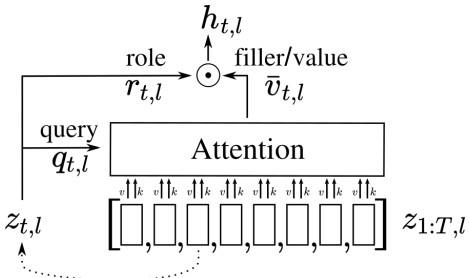

Figure 1: A simplified illustration of our TP-Attention mechanism for one head at position $t$ in layer $l$. The main difference from standard Attention is the additional role representation that is element-wise multiplied with the filler/value representation.

## 2.2 FEED-FORWARD LAYER

The feed-forward layer of a cell consists of an affine transformation followed by a ReLU activation and a second affine transformation:

$$\text{FF}(\boldsymbol{x}) = \boldsymbol{W}_l^{(g)} \text{ReLU}(\boldsymbol{W}_l^{(f)}\boldsymbol{x} + \boldsymbol{b}_l^{(f)}) + \boldsymbol{b}_l^{(g)} \tag{7}$$

Here, $\boldsymbol{W}_l^{(f)} \in \mathbb{R}^{d_f \times d_z}, \boldsymbol{b}_l^{(f)} \in \mathbb{R}^{d_f}, \boldsymbol{W}_l^{(g)} \in \mathbb{R}^{d_z \times d_f}, \boldsymbol{b}_l^{(g)} \in \mathbb{R}^{d_z}$ and $\boldsymbol{x}$ is the function's argument. As in previous work, we set $d_f = 4d_z$.

## 2.3 THE DECODER NETWORK

The decoder network is a separate network with a similar structure to the encoder that takes the hidden states of the encoder and auto-regressively generates the output sequence. In contrast to the encoder network, the cells of the decoder contain two TPMHA layers and one feed-forward layer. We designed our decoder network analogously to Vaswani et al. (2017) where the first attention layer attends over the masked decoder states while the second attention layer attends over the final encoder states. During training, the decoder network receives the shifted targets (teacher-forcing) while during inference we use the previous symbol with highest probability (greedy-decoding). The final symbol probability distribution is given by

$$\hat{\boldsymbol{y}}_{\hat{t}} = \text{softmax}(\boldsymbol{E}^T \hat{\boldsymbol{z}}_{\hat{t},L}) \tag{8}$$

where $\hat{\boldsymbol{z}}_{\hat{t},L}$ is the hidden state of the last layer of the decoder at decoding step $\hat{t}$ of the output sequence and $\boldsymbol{E}$ is the shared symbol embedding of the encoder and decoder.

## 3 THE MATHEMATICS DATASET

The Mathematics Dataset (Saxton et al., 2019) is a large collection of math problems of various types, including algebra, arithmetic, calculus, numerical comparison, measurement, numerical factorization, and probability. Its main goal is to investigate the capability of neural networks to reason formally. Each problem is structured as a character-level sequence-to-sequence problem. The input sequence is a free-form math question or command like `What is the first derivative of 13*a**2 - 627434*a + 11914106?` from which our model correctly predicts the target sequence `26*a - 627434`. Another example from a different module is `Calculate 66.6*12.14.` which has `808.524` as its target sequence.

The dataset is structured into 56 modules which cover a broad spectrum of mathematics up to university level. It is procedurally generated and comes with 2 million pre-generated training samples per module. The authors provide an interpolation dataset for every module, as well as a few extrapolation datasets as an additional measure of algebraic generalization.

We merge the different training splits *train-easy*, *train-medium*, and *train-hard* from all modules into one big training dataset of 120 million unique samples. From this dataset we extract a character-level vocabulary of 72 symbols, including *start-of-sentence*, *end-of-sentence*, and *padding* symbols[2].

## 4 EXPERIMENTAL RESULTS

We evaluate our trained model on the concatenated interpolation and extrapolation datasets of the pre-generated files, achieving a new state of the art: see Table 1. For a more detailed comparison, Fig. 2 shows the interpolation and extrapolation performance of every module separately. The TP-Transformer matches or out-performs the Transformer in every module but one (*probability__swr_p_sequence*). Our model never quite converged, and was stopped prematurely after 1.7 million steps. We trained our model on one server with 4 V100 Nvidia GPUs for 25 days.

Table 1: Model accuracy averaged over all modules. A sample is correct if all characters of the target sequence have been predicted correctly. The column ">95%" counts how many of the 56 modules achieve over 95% accuracy. Boldface marks the best-performing model up to 700k steps.

| | weights | steps | train | interpolation | | extrapolation | |
| --- | --- | --- | --- | --- | --- | --- | --- |
| | | | | acc | >95% | acc | >95% |
| Simple LSTM | 18M | 500k | - | 57.00% | 6 | 41.00% | 1 |
| Transformer (Saxton et al.) | 30M | 500k | - | 76.00% | 13 | 50.00% | 1 |
| Transformer (ours) | 44.2M | 500k | 83.06% | 75.33% | 12 | 52.42% | 1 |
| | | 700k | 85.01% | 77.42% | 14 | 52.00% | 2 |
| TP-Transformer (ours) | 49.2M | 500k | 85.41% | 78.30% | 16 | 52.22% | 2 |
| | | 700k | **87.25%** | **80.67%** | **18** | **52.48%** | **3** |
| | | 1.7M | 91.04% | 84.24% | 25 | 55.40% | 3 |

### 4.1 IMPLEMENTATION DETAILS

We initialize the symbol embedding matrix $\boldsymbol{E}$ from $\mathcal{N}(0,1)$, $\boldsymbol{W}^{(p)}$ from $\mathcal{N}(1,1)$, and all other matrices $\boldsymbol{W}^{(\cdot)}$ using the Xavier uniform initialization as introduced by Glorot & Bengio (2010). The model parameters are set to $d_z = 512, d_f = 2048, d_v = 72, H = 8, L = 6$. We were not able to train the TP-Transformer, nor the regular Transformer, using the learning rate and gradient clipping scheme described by Saxton et al. (2019). Instead we proceed as follows: The gradients are computed using PyTorch's Autograd engine and their gradient norm is clipped at 0.1. The optimizer we use is also Adam, but with a smaller learning_rate $= 1 \times 10^{-4}$, beta1 $= 0.9$, beta2 $= 0.995$. We train with a batch size of 1024 up to 1.7 million steps.

## 5 INTERPRETING THE LEARNED STRUCTURE

We report initial results of analyzing the learned structure of the encoder network's last layer from our 700k-step TP-Transformer.

### 5.1 INTERPRETING THE LEARNED ROLES

To this end, we sample 128 problems from the interpolation dataset of the *arithmetic__mixed* module and collect the role vectors from a randomly chosen head. We use $k$-means with $k = 20$ to cluster the role vectors from different samples and different time steps of the final layer of the encoder. Interestingly, we find separate clusters for digits in the numerator and denominator of fractions. When there is a fraction of fractions we can observe that these assignments are placed such that the second fraction reverses, arguably simplifying the division of fractions into a multiplication of fractions (see Fig. 3).

---

[2]Note that Saxton et al. (2019) report a vocabulary size of 95, but this figure encompasses characters that never appear in the pre-generated training and test data.

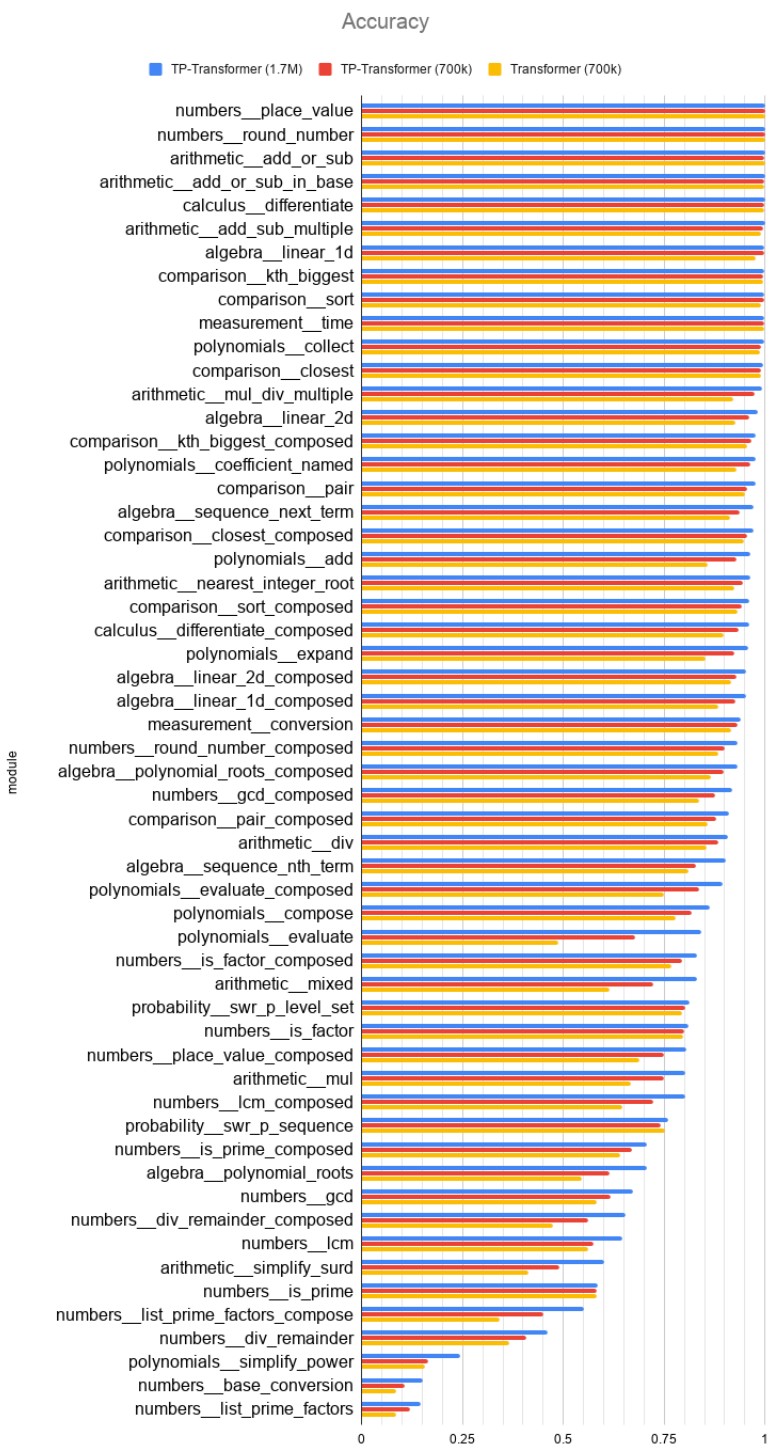

Figure 2: The accuracies of our implementation of the Transformer (700k steps) and the TP-Transformer (700k and 1.7M steps) for every module of the Mathematics Dataset.

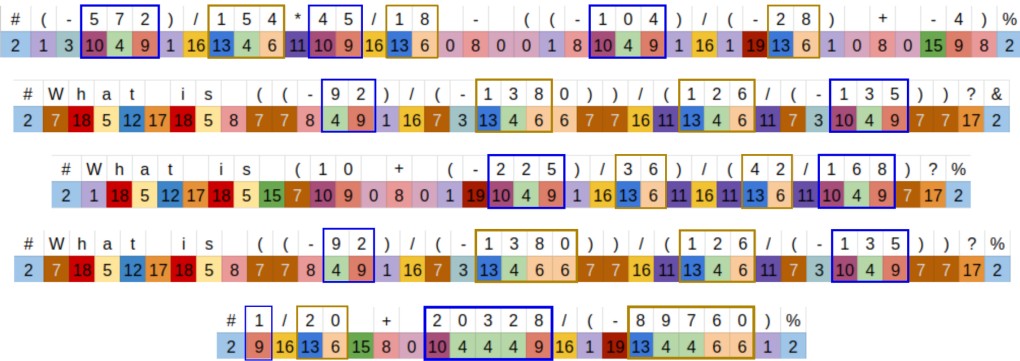

Figure 3: Samples of correctly processed problems from the *arithmetic__mixed* module. '#' and '%' are the start- and end-of-sentence symbols. The colored squares indicate the $k$-means cluster of the role-vector assigned by one head in the final layer in that position. Blue and gold rectangles respectively highlight numerator and denominator roles. They were discovered manually. Note how their placement is correctly swapped in rows 2, 3, and 4, where a number in the denominator of a denominator is treated as if in a numerator. Role-cluster 9 corresponds to the role *ones-digit-of-a-numerator-factor*, and 6 to *ones-digit-of-a-denominator-factor*; other such roles are also evident.

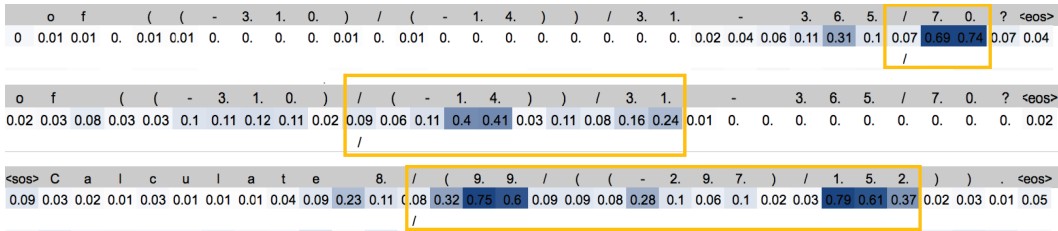

Figure 4: TP-Transformer attention maps for three examples as described in section 5.2.

## 5.2 INTERPRETING THE ATTENTION MAPS

In Fig. 4 we display three separate attention weight vectors of one head of the last TP-Transformer layer of the encoder. Gold boxes are overlaid to highlight most-relevant portions. The row above the attention mask indicates the symbols that take information to the symbol in the bottom row. In each case, they take from '/'. Seen most simply in the first example, this attention can be interpreted as encoding a relation *second-argument-to* holding between the querying digits and the '/' operator. The second and third examples show that several numerals in the denominator can participate in this relation. The third display shows how a numerator-numeral (-297) intervening between two denominator-numerals is skipped for this relation.

## 6 INSIGHTS AND DEFICITS OF MULTIPLE MULTI-HEAD ATTENTION LAYERS

### 6.1 MULTI-HEAD ATTENTION SUBSPACES CAPTURE VIRTUALLY ALL INFORMATION

It was claimed by Vaswani et al. (2017) that *Multi-head attention allows the model to jointly attend to information from different representation subspaces at different positions.* In this section, we show that in our trained models, an individual attention head does not access merely a subset of the information in the attended cell but instead captures nearly the full information content.

Let us consider a toy example where the attention layer of $\text{cell}_{t,l}$ only attends to $\text{cell}_{t',l}$. In this setting, the post-attention representation simplifies and becomes

$$\text{AttentionHead}(\boldsymbol{z}_{t,l}, \boldsymbol{z}_{1:T,l}) = \boldsymbol{z}_{t,l} + \boldsymbol{W}_l^{(o)}(\boldsymbol{W}_l^{(v)}\boldsymbol{z}_{t',l} + \boldsymbol{b}_l^{(v)}) + \boldsymbol{b}_l^{(o)}$$
$$= \boldsymbol{z}_{t,l} + o(v(\boldsymbol{z}_{t',l})) \tag{9}$$

where $o$ and $v$ are the respective affine maps (see Sec. 2.1). Note that even though $\boldsymbol{W}_l^{(v)}$ is a projection into an 8 times smaller vector space, it remains to be seen whether the hidden state loses information about $\boldsymbol{z}_{t',l}$. We empirically test to what extent the trained Transformer and TP-Transformer lose information. To this end, we randomly select $n = 100$ samples and extract the hidden state of the last layer of the encoder $\boldsymbol{z}_{t,6}$, as well as the value representation $\boldsymbol{v}_h(\boldsymbol{z}_{t,6})$ for every head. We then train an affine model to reconstruct $\boldsymbol{z}_{t,6}$ from $\boldsymbol{v}_h(\boldsymbol{z}_{t,6})$, the value vector of the single head $h$:

$$\hat{\boldsymbol{z}}_{t,6} = \boldsymbol{W}_h \boldsymbol{v}_h(\boldsymbol{z}_{t,6}) + \boldsymbol{b}_h$$
$$e = \frac{1}{n}(\hat{\boldsymbol{z}}_{t,6} - \boldsymbol{z}_{t,6})^2 \tag{10}$$

For both trained models, the TP-Transformer and the regular Transformer, the mean squared error $e$ averaged across all heads is only ~0.017 and ~0.009 respectively. This indicates that the attention mechanism incorporates not just a subspace of the states it attends to, but affine transformations of those states that preserve nearly the full information content.

## 6.2 THE BINDING PROBLEM OF STACKED ATTENTION LAYERS

The *binding problem* refers to the problem of binding features together into objects while keeping them separated from other objects. It has been studied in the context of theoretical neuroscience (von der Malsburg, 1981; 1994) but also with regards to connectionist machine learning models (Hinton et al., 1984). The purpose of a binding mechanism is to enable the fully distributed representation of symbolic structure (like a hierarchy of features) which has recently resurfaced as an important direction for neural network research (Lake & Baroni, 2017; Bahdanau et al., 2018; van Steenkiste et al., 2019; Palangi et al., 2017; Tang et al., 2018).

In this section, we describe how the standard attention mechanism is ill suited to capture complex nested representations, and we provide an intuitive understanding of the benefit of our TP-Attention. We understand the attention layer of a cell as the means by which the *subject* (the cell state) queries all other cells for an *object*. We then show how a hierarchical representation of multiple queries becomes ambiguous in multiple standard attention layers.

Consider the string `(a/b)/(c/d)`. A good neural representation captures the hierarchical structure of the string such that it will not be confused with the similar-looking but structurally different string `(a/d)/(c/b)`. Our TP-Attention makes use of a binding mechanism in order to explicitly support complex structural relations by binding together the object representations receiving high attention with a subject-specific role representation. Let us continue with a more technical example. Consider a simplified Transformer network where every cell consists only of a single-head attention layer with a residual connection: no feed-forward layer or layer normalization, and let us assume no bias terms in the maps $o_l$ and $v_l$ introduced in the previous section (Eq. 9). In this setting, assume that $\text{cell}_{a,l}$ only attends to $\text{cell}_{b,l}$, and $\text{cell}_{c,l}$ only attends to $\text{cell}_{d,l}$ where $a, b, c, d$ are distinct positions of the input sequence. In this case

$$\boldsymbol{z}_{a,l+1} = \boldsymbol{z}_{a,l} + o_l(v_l(\boldsymbol{z}_{b,l}))$$
$$\boldsymbol{z}_{c,l+1} = \boldsymbol{z}_{c,l} + o_l(v_l(\boldsymbol{z}_{d,l})) \tag{11}$$

Suppose now that, for hierarchical grouping, the next layer $\text{cell}_{e,l+1}$ attends to both $\text{cell}_{a,l+1}$ and $\text{cell}_{c,l+1}$ (equally, each with attention weight $\frac{1}{2}$). This results in the representation

$$\boldsymbol{z}_{e,l+2} = \boldsymbol{z}_{e,l+1} + o_{l+1}(v_{l+1}(\boldsymbol{z}_{a,l+1} + \boldsymbol{z}_{c,l+1}))/2$$
$$= \boldsymbol{z}_{e,l+1} + o_{l+1}(v_{l+1}(\boldsymbol{z}_{a,l} + \boldsymbol{z}_{c,l} + o_l(v_l(\boldsymbol{z}_{b,l})) + o_l(v_l(\boldsymbol{z}_{d,l}))))/2 \tag{12}$$

Note that the final representation is ambiguous in the sense that it is unclear by looking only at Eq. 12 whether $\text{cell}_{a,l}$ has picked $\text{cell}_{b,l}$ or $\text{cell}_{d,l}$. Either scenario would have led to the same outcome, which means that the network would not be able to distinguish between these two different structures (as in confusing `(a/b)/(c/d)` with `(a/d)/(c/b)`). In order to resolve this ambiguity, the standard Transformer must recruit other attention heads or find suitable non-linear maps in between attention layers, but it remains uncertain how the network might achieve a clean separation.

Our TP-Attention mechanism, on the other hand, specifically removes this ambiguity. Now Eqs. 11 and 12 become:

$$
\begin{aligned}
\boldsymbol{z}_{a,l+1} &= \boldsymbol{z}_{a,l} + o_l(v_l(\boldsymbol{z}_{b,l}) \odot \boldsymbol{r}_{a,l}) \\
\boldsymbol{z}_{c,l+1} &= \boldsymbol{z}_{c,l} + o_l(v_l(\boldsymbol{z}_{d,l}) \odot \boldsymbol{r}_{c,l}) \\
\boldsymbol{z}_{e,l+2} &= \boldsymbol{z}_{e,l+1} + o_{l+1}(v_{l+1}(\boldsymbol{z}_{a,l} + \boldsymbol{z}_{c,l} + o_l(v_l(\boldsymbol{z}_{b,l}) \odot \boldsymbol{r}_{a,l}) + o_l(v_l(\boldsymbol{z}_{d,l}) \odot \boldsymbol{r}_{c,l})))/2
\end{aligned}
\tag{13}
$$

Note that the final representation is not ambiguous anymore. Binding the filler symbols $v_l(\boldsymbol{z})$ (our objects) with a subject-specific role representation $\boldsymbol{r}$ as described in Eq. 6 breaks the structural symmetry we had with regular attention. It is now simple for the network to specifically distinguish the two different structures.

## 7  RELATED WORK

Several recent studies have shown that the Transformer-based language model BERT (Devlin et al., 2018) captures linguistic relations such as those expressed in dependency-parse trees. This was shown for BERT's hidden activation states (Hewitt & Manning, 2019; Tenney et al., 2019) and, most directly related to the present work, for the graph implicit in BERT's attention weights (Coenen et al., 2019; Lin et al., 2019). Future work applying the TP-Transformer to language tasks (like those on which BERT is trained) will enable us to study the connection between the *explicit* relations $\{\boldsymbol{r}_{t,l}^h\}$ the TP-Transformer learns and the *implicit* relations that have been extracted from BERT.

## 8  CONCLUSION

We have introduced the TP-Transformer, which enables the powerful Transformer architecture to learn to explicitly encode structural relations using Tensor-Product Representations. On the novel and challenging Mathematics Dataset, TP-Transformer beats the previously published state of the art by 8.24%. Our initial analysis of this model's final layer suggests that the TP-Transformer naturally learns to cluster symbol representations based on their structural position and relation to other symbols.

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

# A    APPENDIX: RELATIONS BETWEEN HADAMARD- AND TENSOR-PRODUCT-BINDING

## A.1    GENERAL CONSIDERATIONS

In the version of the TP-Transformer studied in this paper, binding of relations $r$ to their values $v$ is not done by the tensor product, $v \otimes r$, as in full TPRs. Rather, a contraction of the full TPR is used: the diagonal, which is the elementwise or Hadamard product $v \odot r$.[3]  To what extent does Hadamard-product binding share relevant properties with tensor-product binding?

A crucial property of the tensor product for its use in vector representations of structure is that a structure like a/b is not confusable with b/a, unlike the frequently-used bag-of-words encoding: in the BOW encoding of a/b, the pair of arguments to the operator are encoded simply as $a + b$, where $a$ and $b$ are respectively the vector encodings of a and b. Obviously, this cannot be distinguished from the BOW encoding of the argument pair in b/a, $b + a$. (Hence the name, symbol "bag", as opposed to symbol "structure".)

In a tensor-product representation of the argument pair in a/b, we have $a \star r_n + b \star r_d$, where $r_n$ and $r_d$ are respectively distinct vector embeddings of the numerator (or first-argument) and denominator (or second-argument) roles, and $\star$ is the tensor product. This is distinct from $a \star r_d + b \star r_n$, the embedding of the argument-pair in b/a. (In Sec. 6.2 of the paper, an aspect of this general property, in the context of attention models, is discussed. In Sec. 5, visualization of the roles and the per-role-attention show that this particular distinction, between the numerator and denominator roles, is learned and used by the trained TP-Transformer model.)

This crucial property of the tensor product, that $a \star r_n + b \star r_d \neq a \star r_d + b \star r_n$, is shared by the Hadamard product: if we now take $\star$ to represent the Hadamard product, the inequality remains true. To achieve this important property, the full tensor product is not required: the Hadamard product is the diagonal of the tensor product, which retains much of the product structure of the tensor product. In any application, it is an empirical question how much of the full tensor product is required to successfully encode distinctions between bindings of symbols to roles; in the TP-Transformer, it turns out that the diagonal of the tensor product is sufficient to get improvement in performance over having no symbol-role-product structure at all. Unfortunately, the compute requirements of training on the Mathematics Dataset currently makes using the full tensor product infeasible, unless the vector representations of symbols and roles are reduced to dimensions that proved to be too small for the task. When future compute makes it possible, we expect that expanding from the diagonal to the full tensor product will provide further improvement in performance and interpretability.

We next move beyond these general considerations and consider a setting in which Hadamard-product attention can yield an optimal approximation to tensor-product attention.

## A.2    HADAMARD-PRODUCT ATTENTION AS AN OPTIMAL APPROXIMATION TO TENSOR-PRODUCT ATTENTION

In Eq. 6 for TPMHA, we have a sum over all $H$ heads of an affine-transformed product of a value vector $v_h$ and a role vector $r_h$. (Throughout this discussion, we leave the subscripts $t, l$ implicit, as well as the over-bar on $v_h$ in Eq. 6.) In a hypothetical, full-TPR formulation, this product would be the tensor product $v_h \otimes r_h$, although in our actual proposed TP-Transformer, the Hadamard (elementwise) product $v_h \odot r_h$ (the diagonal of $v_h \otimes r_h$) is used. The appropriateness of the compression from tensor product to Hadamard product can be seen as follows.

In the hypothetical full-TPR version of TPMHA, attention would return the sum of $H$ tensor products. This tensor **A** would have rank at most $H$, potentially enabling a substantial degree of compression across all tensors the model will compute over the data of interest. Given the translation-invariance built into the Transformer via position-invariant parameters, the same compression must be applied in all positions within a given layer $l$, although the compression may vary across heads. For the compression of **A** we will need more than $H$ components, as this decomposition needs to be optimal over all *all* tensors in that layer for *all* data points.

---

[3]This is a vector, and should not be confused with the inner product $v \cdot r$ which is a scalar: the inner product is the sum of all the elements of the Hadamard product.

In detail, for each head $h$, the compression of the tensor $\mathbf{A}_h = \boldsymbol{v}_h \otimes \boldsymbol{r}_h$ (or matrix $\boldsymbol{A}_h = \boldsymbol{v}_h \boldsymbol{r}_h^\top$) is to dimension $d_k$, which will ultimately be mapped to dimension $d_z$ (to enable addition with $\boldsymbol{z}$ via the residual connection of Eq. 1) by the affine transformation of Eq. 6 . The optimal $d_k$-dimensional compression for head $h$ at layer $l$ would preserve the $d_k$ dominant dimensions of variance of the attention-generated states for that head and layer, across all positions and inputs: a kind of singular-value decomposition retaining those dimensions with the principal singular values. Denote these principal directions by $\{\boldsymbol{m}_c^h \otimes \boldsymbol{n}_c^h | c = 1, \ldots, d_k\}$, and let $\boldsymbol{M}_h$ and $\boldsymbol{N}_h$ respectively be the $d_k \times d_k$ matrices with the orthonormal vectors $\{\boldsymbol{m}_c^h\}$ and $\{\boldsymbol{n}_c^h\}$ as columns. (Note that orthonormality implies that $\boldsymbol{M}_h^\top \boldsymbol{M}_h = \boldsymbol{I}$ and $\boldsymbol{N}_h^\top \boldsymbol{N}_h = \boldsymbol{I}$, with $\boldsymbol{I}$ the $d_k \times d_k$ identity matrix.)

The compression of $\mathbf{A}_h$, $\hat{\mathbf{A}}_h$, will lie within the space spanned by these $d_k$ tensor products $\boldsymbol{m}_c^h \otimes \boldsymbol{n}_c^h$, i.e., $\hat{\mathbf{A}}_h = \sum_{c=1}^{d_k} \tilde{a}_c^h \boldsymbol{m}_c \otimes \boldsymbol{n}_c$; in matrix form, $\hat{\boldsymbol{A}}_h = \boldsymbol{M}_h \tilde{\boldsymbol{A}}_h \boldsymbol{N}_h^\top$, where $\tilde{\boldsymbol{A}}_h$ is the $d_k \times d_k$ diagonal matrix with elements $\tilde{a}_c^h$. Thus the $d_k$ dimensions $\{\tilde{a}_c^h\}$ of the compressed matrix $\hat{\boldsymbol{A}}_h$ that approximates $\boldsymbol{A}_h$ are given by:

$$\tilde{\boldsymbol{A}}_h = \boldsymbol{M}_h^\top \hat{\boldsymbol{A}}_h \boldsymbol{N}_h \approx \boldsymbol{M}_h^\top \boldsymbol{A}_h \boldsymbol{N}_h = \boldsymbol{M}_h^\top \boldsymbol{v}_h \boldsymbol{r}_h^\top \boldsymbol{N}_h = (\boldsymbol{M}_h^\top \boldsymbol{v}_h)(\boldsymbol{N}_h^\top \boldsymbol{r}_h)^\top$$

$$\tilde{a}_c^h = \left[\tilde{\boldsymbol{A}}_h\right]_{cc} \approx [\boldsymbol{M}_h^\top \boldsymbol{v}_h]_c [\boldsymbol{N}_h^\top \boldsymbol{r}_h]_c = [\tilde{\boldsymbol{v}}_h \odot \tilde{\boldsymbol{r}}_h]_c$$

where $\tilde{\boldsymbol{v}}_h = \boldsymbol{M}_h^\top \boldsymbol{v}_h$, $\tilde{\boldsymbol{r}}_h = \boldsymbol{N}_h^\top \boldsymbol{r}_h$. Now from Eq. 3, $\boldsymbol{v}_h = \boldsymbol{W}^{h,(v)} \boldsymbol{z} + \boldsymbol{b}^{h,(v)}$, so $\tilde{\boldsymbol{v}}_h = \boldsymbol{M}_h^\top (\boldsymbol{W}^{h,(v)} \boldsymbol{z} + \boldsymbol{b}^{h,(v)})$. Thus by changing the parameters $\boldsymbol{W}^{h,(v)}$, $\boldsymbol{b}^{h,(v)}$ to $\widetilde{\boldsymbol{W}}^{h,(v)} = \boldsymbol{M}_h^\top \boldsymbol{W}^{h,(v)}$, $\tilde{\boldsymbol{b}}^{h,(v)} = \boldsymbol{M}_h^\top \boldsymbol{b}^{h,(v)}$, and analogously for the role parameters $\boldsymbol{W}^{h,(r)}$, $\boldsymbol{b}^{h,(r)}$, we convert our original hypothetical TPR attention tensor $\mathbf{A}_h$ to its optimal $d_k$-dimensional approximation, in which the tensor product of the original vectors $\boldsymbol{v}_h$, $\boldsymbol{r}_h$ is replaced by the Hadamard product of the linearly-transformed vectors $\tilde{\boldsymbol{v}}_h$, $\tilde{\boldsymbol{r}}_h$. Therefore, in the proposed model, which deploys the Hadamard product, learning simply needs to converge to the parameters $\widetilde{\boldsymbol{W}}^{h,(v)}$, $\tilde{\boldsymbol{b}}^{h,(v)}$ rather than the parameters $\boldsymbol{W}^{h,(v)}$, $\boldsymbol{b}^{h,(v)}$.

## A.3 NEURAL MACHINE TRANSLATION

To demonstrate the generalizability of our contribution, we run a preliminary neural machine translation experiment. We follow the experimental setup of the fairseq toolkit (Ott et al., 2019). The dataset was modeled as in previous work (Vaswani et al., 2017). The dataset consists of the WMT'14 English to German data but adds additional news-commentary-v12 data from WMT'17. We use byte-pair encoding and employed label-smoothing with a value of 0.1. We train the regular transformer and our TP-Transformer on 4 P100 GPUs for 500,000 steps using the hyperparameters of the regular transformer. Throughout training, we observe that our TP-Transformer achieves a lower negative log-likelihood than the regular transformer.

Table 2: The results of our neural machine translation experiment. The loss is the smoothed negative log-likelihood of the word-pieces (lower is better). The TP-Transformer achieves a lower loss throughout training and a comparable final BLEU score (higher is better).

|  | loss | BLEU |
| --- | --- | --- |
| Transformer (ours) | 4.137 | 27.05 |
| TP-Transformer (ours) | **4.111** | **27.10** |

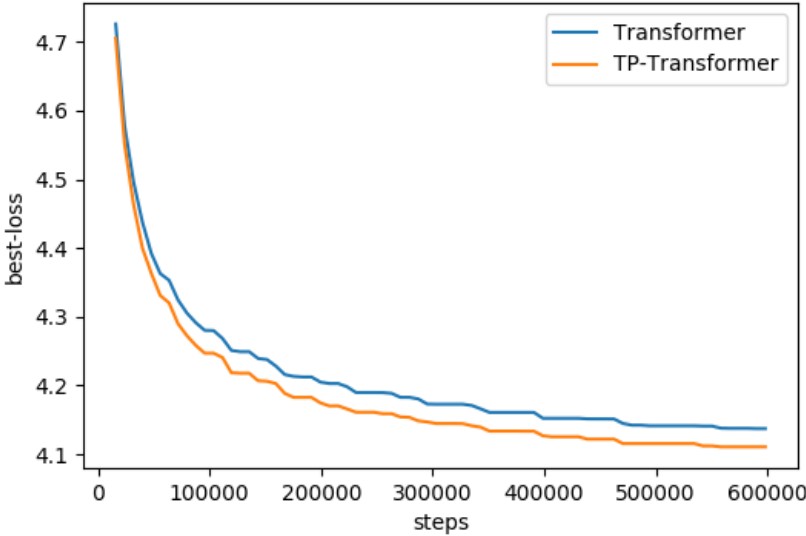

Figure 5: The best-so-far smoothed negative log-likelihood of word-pieces on held-out data through-out training.

