# OpenReview forum: "Enhancing the Transformer with explicit relational encoding for math problem solving"
_ICLR.cc/2020/Conference — Reject_

### Official Review · AnonReviewer2 · 2019-10-22
**Official Blind Review #2**

**Rating:** 6

**Review:**

This paper illustrates the TP-Transformer architecture on the challenging mathematics dataset. The TP-Transformer combines the transformer architecture with tensor-product representations. The experiments show a dramatic improvement of accuracies compared with SOTA models. Moreover, the paper also explains the reason why the TP-Transformer can learn the structural position and relation to other symbols with a detailed math proof.

Overall, this paper is nice as it makes a milestone for math problem solving from unique perspectives. To be specific, the paper makes the following contributions:

1. Demonstrate a novel architecture TP-Transformer in details;
2. Achieve a better accuracies in the challenging mathematics dataset than the SOTA transformer models;
3. Illustrate in fundamental math that why TP-Transformer can learn the structural position and relation, and solve the binding problems of stacked attention layers.

Here are a few minor questions that may further improve the paper:

1. The conclusion states that TP-Transformer beats the previously published SOTA by 8.24%. However, it does not match to the experiment results (see section 4).

2. In figure 5, there are 4 tasks in the bottom with accuracies lower than 0.5. It would be nice to provide more insights on this.

3. It would be interesting to see whether it transferable to the other downstream tasks (such as natural language understanding) besides the experiments on the challenging mathematics dataset.

**Experience Assessment:**

I have published in this field for several years.

**Review Assessment: Checking Correctness Of Derivations And Theory:**

I carefully checked the derivations and theory.

**Review Assessment: Checking Correctness Of Experiments:**

I carefully checked the experiments.

**Review Assessment: Thoroughness In Paper Reading:**

I read the paper thoroughly.

---

> ### Author Response · Authors · 2019-11-11
> **Response to AnonReviewer2**
>
> Thank you for your enthusiastic review. We appreciate that you share our excitement with regards to our progress on this extremely challenging dataset. One of the great features of this dataset is the opportunity to investigate deeply compositional problems. Due to its careful creation and design, it does not suffer from good-performing but shallow heuristics as is often the case in natural-language-based datasets. This makes it a great testbed to develop models that can represent and reason over more symbolic structures. We are happy to answer your questions:
>
> 1.) Please see point 1 of our response to AnonReviewer3 who raised a very similar question. Please let us know if you then still consider the conclusion misleading as it is.
>
> 2.) The dataset is designed such that the answers are virtually impossible to guess. For an answer to be correct, all characters of the answer sequence must be predicted accurately. We observed that our trained model often only gets a single character wrong which renders the whole answer wrong. This results in a performance drop throughout all tasks -- we did not analyse this in detail. The four tasks on which our model performs worst are tasks such as generating the prime factors of a number. Consider this character sequence input: "What are the prime factors of 2104900?" from which the TP-Transformer has to predict the sequence "2, 5, 7, 31, 97". The other tasks are similar in difficulty; this is best explained through an actual sample.
>
> div_remainder: "What is the remainder when 61720 is divided by 183?" Answer: "49"
> simplify_power: "Simplify n**(-31)*n**(2/73)*(n/(n/(n/n**(1/3))))/n**(2/19)*n**21 assuming n is positive." Answer: "n**(-39160/4161)"
>
> We added an anonymous colab notebook enabling you to experiment on your own in your browser. You’ll find the link in our general comment which summarizes all our revisions.
>
> 3.) We followed your advice and included a neural machine translation experiment. Throughout training our TP-Transformer model achieves a lower negative log-likelihood when compared to the regular Transformer with the same hyper-parameters.
>
> We hope you find our modifications sufficient for the revised version of this paper to be accepted, and welcome any further questions or suggestions.

---

### Official Review · AnonReviewer3 · 2019-10-24
**Official Blind Review #3**

**Rating:** 6

**Review:**

In this paper, the authors incorporated tensor-product representations within the Transformer.  By creating an attention mechanism called TP-Attention, they explicitly encode the relations between each Transformer cell and the other cells, whose values are retrieved by attention. By introducing tensor products, the proposed algorithm can empirically perform well for noncommutative operations with multiple arguments, such as division. The authors trained models with the proposed algorithm on the Mathematics Dataset and compared the performances with two baselines (simple LSTM and the original Transformer). At last, several model snapshots are provided to help interpret several key elements of the model: the learned roles, the attention maps, the TP-transformer columns and so on.

Overall, the paper is well-written. The experimental results generally support the high-level intuition behind the introduction of tensor-product representation. I would recommend accepting this paper.

Some quick questions:

1. It was claimed in the Conclusion section that the performance of the proposed algorithm beats the previously published state of the art by 8.24%. I guess the number comes from the 2nd and the last row of interpolation accuracy in Table 1. However, these two results are obviously trained for different numbers of iterations: The baseline algorithm was trained for 500k steps, while the proposed algorithm is trained for 1.7M steps. Is it a fair comparison? If the proposed algorithm is also trained for 500k steps, the improvement is around 2.3%.

2. Why is the extrapolation accuracy results for TP-Transformer missing in Table 1?



**Experience Assessment:**

I do not know much about this area.

**Review Assessment: Checking Correctness Of Derivations And Theory:**

N/A

**Review Assessment: Checking Correctness Of Experiments:**

I assessed the sensibility of the experiments.

**Review Assessment: Thoroughness In Paper Reading:**

I made a quick assessment of this paper.

---

> ### Author Response · Authors · 2019-11-11
> **Response to AnonReviewer3**
>
> Thank you for your review and your questions. We are pleased to hear that you appreciated our manuscript and consider it well written.
>
> 1.) Your first question is understandable. In section 4 of our manuscript, we provide a controlled comparison to justify our modifications to the Transformer architecture. Note the boldface on the 700k step version. However, the claim of a new state of the art in our conclusion is different, as it does not assert the source of the improvement. Here, as done throughout the literature, a claim of a new state of the art with x% improvement is a simple comparison of the best results before vs after our work. Such claims are routinely made when the "before" and "after" models differ substantially in training data and time and parameter count. That said, we are willing to change this statement if the reviewers consider it misleading.
>
> 2.) This result is no longer missing.
>
> We understand that you are not entirely familiar with this line of work. Nevertheless, we hope that you find our arguments convincing enough to accept the revised version. We are happy to answer any further questions you might have.

---

### Official Review · AnonReviewer1 · 2019-10-24
**Official Blind Review #1**

**Rating:** 3

**Review:**

Motivated by the fact that the attention mechanism in transformers is symmetric which might not be able to disambiguate different orders, this work proposes to use a subject vector (in addition to query, key states) for each attention head, and multiply it elementwise with the context vector for each head before merging the heads. Experiments on a mathematics dataset shows superior performance compared to the normal transformer. Qualitatively, the proposed model exhibits attentions that are more interpretable, and clustering by the subject vector gives some insights into how the model solved this problem.

Pros:
1. This work shows better performance than baseline transformer.
2. The clustering of the subject vectors gives some insights into model's behavior .

Cons:
1. In terms of experiments, the proposed approach adds a few million parameters to normal transformer (table 1), but in terms of interpolation it only improves 3% (extrapolation improves 0.5%) at 700k steps. The comparison would be fairer if the normal transformer can be given more parameters.
2. In terms of experiments, this approach is only evaluated on the mathematics dataset, but the argument for relational encoding is pretty general. It would be nice if experiments on other tasks are shown in addition to the math dataset.
3. In terms of motivation, the claim that there're ambiguities introduced by multiple layers of  regular attention needs to be supported by evidence. I think (which authors also pointed out) the feedforward network and non-linearties can disambiguate as well.
4. In terms of interpretablity, there's claim that the learned attention maps more interpretable than transformer. Can there be more quantitative measures? It appears to me that both are hard to interpret.

While this work shows superior performance on the mathematics dataset, I have a few concerns about the generalizability of this proposed architectural change to other problems, as well as the fairness of comparison to baseline. Therefore, I am inclined to reject this paper.

----updates after reading rebuttal----
Thanks for adding the new NMT experiment in Appendix A3. My concern is that the proposed TP-Transformer is not very effective on NMT. Therefore, I'm keeping my score.

**Experience Assessment:**

I have published in this field for several years.

**Review Assessment: Checking Correctness Of Derivations And Theory:**

I carefully checked the derivations and theory.

**Review Assessment: Checking Correctness Of Experiments:**

I carefully checked the experiments.

**Review Assessment: Thoroughness In Paper Reading:**

I read the paper thoroughly.

---

> ### Author Response · Authors · 2019-11-11
> **Response to AnonReviewer1**
>
> Thank you for your detailed review.
>
> Please note that our motivation is to better incorporate structure into the hidden representations of the transformer. We propose a relation vector to explicitly label the information-flow from one cell to another. We argue in the manuscript that such a representational space is best represented as a Tensor Product Representation (TPR). Unfortunately, a full Tensor Product is not feasible for models of this size which is why we instead make use of an optimal approximation: the Hadamard product of the relation vector and the attention-weighted value vectors. We added two additional sections to the appendix that justify our decisions in great detail. We’d appreciate your feedback on those.
>
> We'll now address your comments:
>
> 1. We considered fairness in the sense that both architectures have the same hidden state size. For this type of model, it is not possible to satisfy this constraint and equal number of parameters reasonably. Also, keeping the hidden state size the same maintains the closest match to the regular transformer model.
>
> 2. We followed your advice and added a preliminary Machine Translation experiment to the appendix of the manuscript. In our experiment, our TP-Transformer achieved a lower negative log-likelihood compared to the regular Transformer.
>
> 3. As you have pointed out: this section is merely a theoretical exercise that highlights the drawbacks of multiple layers of attention. We want to stress that this is not our main motivation but a closely related insight. Even though a neural network with a non-linear activation function is a general function approximator, that doesn't mean that the network is likely going to learn to disambiguate representations in the right way. Therefore, we argue to incorporate a simple inductive bias in the form of our TPR approximation between the relation vector and the linear combination of value vectors.
>
> 4. This is a valid concern. We agree and decided to defer a detailed analysis of the learned structure for future work. We removed this section from our manuscript.
>
> We believe that the arguments and experiments are now sufficient evidence for the generalizability of our contribution. In future work, we plan to apply the TP-Transformer to language modelling, and we believe our empirical and theoretical results so far give us good reasons to believe that this will be a promising direction.
>
> We are looking forward to your response.

---

### Public Comment · ~Hyunjae_Kim1 · 2019-10-05
**tensor-product  ?**

Thank you for the interesting work.

You highlighted the tensor-product operation in the introduction of the paper,
but in practice, the inner product was used.

I wonder if any mathematical property of the tensor-product was used in the paper.

---

> ### Author Response · Authors · 2019-10-10
> **We use the tensor product property of the Hadamard product. Not the inner product.**
>
> Thank you for your comment. What is used in the version of the TP-Transformer studied in this paper is not actually the inner product, which returns a single scalar, but the Hadamard or element-wise product, which returns a vector: there is no summation of products in the Hadamard product as there is in the inner product.
>
> A crucial property of the tensor product for its use in vector representations of structure is that a structure like a/b is not confusable with b/a, unlike the frequently-used bag-of-words encoding: in the BOW encoding of a/b, the pair of arguments to the operator are encoded as A + B, where A and B are the vector encodings of a and b respectively. Obviously, this cannot be distinguished from the BOW encoding of the argument pair in b/a, B + A. (Hence the name symbol “bag”, as opposed to symbol “structure”.)
>
> In a tensor-product representation of the argument pair in a/b, we have A * N + B * D, where N and D are respectively distinct vector embeddings of the numerator (or first-argument) and denominator (or second-argument) roles, and * denotes the tensor product. This is distinct from A * D + B * N, the embedding of the argument-pair in b/a. (In Sec. 6.2 of the paper, an aspect of this general property, in the context of attention models, is discussed. In Sec. 5, visualization of the roles and the per-role-attention show that this particular distinction, between the numerator and denominator roles, is learned and used by the trained TP-Transformer model.)
>
> This crucial property of the tensor product, that  A * N + B * D =! A * D + B * N, is shared by the Hadamard product, so if we now take * to represent the Hadamard product, the inequality remains true. To achieve this important property, the full tensor product is not required: the Hadamard product is the diagonal of the tensor product, which retains much of the product structure of the tensor product. In any application, it is an empirical question how much of the full tensor product is required to successfully encode distinctions between bindings of symbols to roles; in the TP-Transformer, it turns out that the diagonal of the tensor product is sufficient to get improvement in performance over having no symbol-role-product structure at all. Unfortunately, the compute requirements of training on the Mathematics Dataset made using the full tensor product infeasible, unless the vector representations of symbols and roles were reduced to dimensions that proved to be too small for the task. When future compute makes it possible, we expect that expanding from the diagonal to the full tensor product will provide further improvement in performance and interpretability.

---

### Author Response · Authors · 2019-11-11
**Summary of Revision**

Dear Reviewers,

Thank you for your valuable time. We have revised our manuscript. We will summarize our changes below and address the details of each review directly.

1.) We have improved the readability of our text and equations throughout the manuscript.
2.) We have added the previously missing extrapolation results for our 500k step TP-Transformer model (as pointed out by R3)
3.) We made sure the comparison in Figure 2 is fair by adding the per-task performances of the 700k step TP-Transformer (this was a concern raised by R1).
4.) We have removed section 6 "Interpretation of a TP-Transformer column". We believe that an analysis of the attention weights of our TP-Transformer is valuable if done adequately. During our revision, we decided that this section not on par with the quality of the other sections. As such, we decided to leave such an analysis for future work. R3 had a similar comment.
5.) We have added two sections to the Appendix to clarify the relations between the Hadamard- and Tensor-Product-Binding (A.1) and the appropriateness of the Hadamard product as a compression of the Tensor product (A.2). We kindly ask the reviewers to examine these two additional sections. They address comments made by all reviewers.
6.) We have added an experiment to the appendix (A.3) to attest the generality of the TP-Attention. We compare the regular Transformer and our TP-Transformer (using the hyperparameters of the regular Transformer) on the WMT’14 en-de translation dataset with additional data from WMT’17. Throughout training we find that the TP-Transformer achieves a lower negative log-likelihood when compared with the regular Transformer. This addresses comments made by R1 and R2.

Additionally, we’d like to announce the following:

7.) We have prepared an anonymous Google colab notebook for anyone to experiment with our best TP-Transformer: https://colab.research.google.com/drive/1hXUmTXkN2mXF07mcv9uim14BhjpPpqiY
8.) We have prepared our source code, pretrained models, and preprocessed data for publication. We’ll include the respective links to the final deanonymized version of this manuscript.

---

### Decision · Program_Chairs · 2019-12-19

**Decision:**

Reject

**Comment:**

This paper proposes a change in the attention mechanism of Transformers yielding the so-called "Tensor-Product Transformer" (TP-Transformer). The main idea is to capture filler-role relationships by incorporating a Hadamard product of each value vector representation (after attention) with a relation vector, for every attention head at every layer. The resulting model achieves SOTA on the Mathematics Dataset. Attention maps are shown in the analysis to give insights into how TP-Transformer is capable of solving the Mathematics Dataset's challenging problems.

While the modified attention mechanism is interesting and the analysis is insightful (and improved with the addition of an experiment in NMT after the rebuttal), the reviewers expressed some concerns in the discussion stage:

1. The comparison to baseline is not fair (not to mention the 8.24% claim in conclusion). The proposed approach adds 5 million parameters to a normal transformer (table 1, 5M is a lot!), but in terms of interpolation, it only improves 3% (extrapolation improves 0.5%) at 700k steps. The rebuttal claimed that it is fair as long as the hidden size is comparable, but I don't think that's a fair argument. I suspect that increasing the feedforward hidden size (d_ff) of a normal transformer to match parameters (and add #training steps to match #train steps) might change the conclusion.
2. The new experiment on WMT further convinces me that the theoretical motivation does not hold in practice. Even with the added few million more parameters, it only improved BLEU by 0.05 (we usually consider >0.5 as significant or non-random). This might be because the feedforward and non-linearity can disambiguate as well.

I also found the name TP-Transformer a bit misleading, since what is proposed and tested here is the Hadamard product (i.e. only the diagonal part of the tensor product).

I recommend resubmitting an improved version of this paper with  stronger empirical evidence of outperformance of regular Transformers with comparable number of parameters.